# Community-Led Health Initiatives for Torres Straits Island Communities in a Changing Climate: Implementing Core Values for Mitigation and Adaptation

**DOI:** 10.3390/ijerph192416574

**Published:** 2022-12-09

**Authors:** Nina Lansbury, Andrew M. Redmond, Francis Nona

**Affiliations:** 1School of Public Health, The University of Queensland, Herston, QLD 4006, Australia; 2Royal Brisbane and Women’s Hospital, Herston, QLD 4029, Australia; 3Faculty of Medicine, The University of Queensland, Herston, QLD 4006, Australia

**Keywords:** First Nations, indigenous, Torres Strait Islands, resilience, community participation, social determinants, social and environmental justice, climate change, health and wellbeing

## Abstract

First Nations Peoples have a long history of living in Australia’s changing climate and a deep knowledge of their traditional estate (‘Country’). However, human-induced climate change raises unforeseen risks to the health of First Nations Peoples—especially in remotely located communities. This includes the Torres Strait Islands, where a local leader asked our Torres Strait Islander co-author, ’We know that you will return to your Country—unlike previous researchers. So how can you help with climate change?’ In response, this research describes four core values focused on supporting First Nations Peoples’ health and wellbeing: co-design, appropriate governance, support for self-determination, and respectfully incorporating Indigenous Knowledges into health-protective climate initiatives. Supporting the health and wellbeing of Torres Strait Islanders to continue living in the remote Torres Strait Islands in a changing climate can enable long-term care for Country, maintenance of culture, and a sense of identity for First Nations Peoples. Ensuring these core values are implemented can support the health of present and future generations and will likely be applicable to other First Nations communities.

## 1. Introduction


*As a Torres Strait Islander, I am especially interested in the impacts of climate change on the health of the people in the Torres Straits—including the social and emotional impacts to the people of the region that stem from disruption to the environment… I am genetically, personally, and spiritually connected to the lands and seas of this region.*


*While it is important to me as a Torres Strait Islander man, the effects of climate change are issues that affect us globally—social justice, wellbeing, economic issues. By their geographical nature, the Torres Straits are particularly vulnerable and are already suffering the effects of climate change. Ongoing climate change threatens to permanently change parts of my homeland forever*.[1]

This statement was delivered by a co-author of this paper, Francis Nona, on one of many occasions when he has been invited to discuss current and pending impacts of climate change on his homeland (‘Country’) in the remote Torres Strait Islands of Queensland, Australia.

There are several embedded assumptions within the work that Francis Nona undertakes in multidisciplinary research groups on this topic with both non-Indigenous allies and fellow First Nations Australians (the term used in this paper to respectfully refer to Aboriginal and Torres Strait Islander Peoples) [2,3]. Firstly, there is a recognition and respect for First Nations Australians to live on ‘Country’ (traditional estate) as the traditional custodians of these lands and as expression of their human rights. This includes the rights specifically outlined in the United Nations (UN) Declaration on the Rights of Indigenous Peoples [4,5]. Secondly, living on Country can be supportive of the social, emotional, spiritual, and physical wellbeing of the traditional custodians [6,7,8]. This definition of health and wellbeing is held by many First Nations Peoples as being more holistic and culturally appropriate than the narrower biomedical model that focuses on ill-health as the basis for treatment [9,10]. Thirdly, Australia’s First Nations Peoples have deep knowledge of their traditional Country and a long history of living in a changing climate [11,12,13]. Therefore, Indigenous Knowledges held by custodians regarding managing and caring for Country provide crucial and geographically specific insights for both mitigation and adaptation of the current climate era [14,15,16].

Despite this strong foundation to respond to climate change, the ongoing impacts of colonisation—notably poverty, institutional and other racism, and land possession—can challenge the ability to adapt to living in a changing climate [14]. Furthermore, health vulnerabilities of community members can be exacerbated where Country is remote from urban centres and the associated protective services, infrastructure, and political attention [2,17,18]. In combination, this can position First Nations Australians in remote communities as priority populations for support with climate change responses [19].

Climate change impacts on the health of remote-living First Nations communities have not previously been extensively documented in academic and policy publications (for example, see [15,20]. A lack of documentation can render the climate change impacts on remote populations as ‘unseen’ by policy and funding agencies [21,22]. Therefore, collecting and publishing climate impact data and the lived experiences of remote communities is important to engender attention and subsequent action.

However, the researchers who collect and publish such data need to develop rapport and demonstrate trust in partnership with such communities [23]. This approach prevents the continuation of an historical pattern of data being collected without adequate community consent and transparency [24,25]. Instead, the research process would best occur where First Nations researchers play a prominent and guiding role and cultural awareness is demanded of non-Indigenous allies involved in that research [26]. Such an approach was requested on a research relationship visit in 2021 to Waiben/Thursday Island, where Francis Nona was invited by a Torres Strait Islander community leader to take a leadership role in climate change research and responses for the region. He was asked:


*We know that you will return to your Country—unlike previous researchers. So how can you help with climate change?*
(Torres Strait leader, personal communication, 22 September 2021).

In responding to this question and invitation from his community living on the Torres Strait Islands and within the wider Torres Strait Islander diaspora [27], this paper seeks to describe the core values of relevance to provide First Nations-led guidance for climate change initiatives. While described through a focus on the Torres Strait, it is anticipated that these values can be adapted beyond the Torres Strait region and communities to climate and health responses where First Nations Peoples and Country are affected.

## 2. Methods

This climate change-focused paper was developed during a time of climate policy change in Australia. In September 2022, the Australian Government passed the Climate Change Bill 2022, with the stated object to ‘to advance an effective and progressive response to the urgent threat of climate change’ ([28] s3 (aa)). Of direct relevance to this research and the health impacts of climate change to the Torres Strait Islander communities was the direction under this Act for the responsible Minister to submit an annual climate change statement that considers, among other aspects, the ‘social, employment and economic benefits being delivered by those policies in rural and regional Australia’ ([28] s12 (e)) and ‘risks to Australia from climate change impacts, such as those relating to Australia’s environment, biodiversity, health…’ ([28] s12 (f)).

Within this policy context, this paper sought to synthesise a range of published evidence on authentic and respectful cultural engagement in First Nations communities, on the observed impacts of climate change, and on the impacts on health from climate change—delivered through an on-ground focus on the Torres Strait Islands. This is in contrast to previous publications that have either provided only a high-level overview of climate change impacts on health for remote communities [17,29] or not included First Nations authorship from those Countries [30,31]. Although these overview publications have provided a helpful synthesis, the lack of specificity and impact on cultural heritage and specific geographies are lacking, as well as the guiding First Nations voice. Both these absences were addressed in this research through an applied focus on the Torres Strait Islands with First Nations research leadership.

This paper was prepared under the guidance of our Torres Strait Islander co-author of the writing collaboration, Francis Nona, with two non-Indigenous allies, Nina Lansbury and Andrew Redmond. Francis Nona’s cultural leadership, in-community engagement on his Country, and lived experience were integral to ensuring an appropriate focus, cultural competency, and portrayal were undertaken [27]. His placement as final author follows the medical discipline where first and last author positions are traditionally ranked as most important [32]. On this paper, Francis Nona is recognised for his seniority in terms of cultural understanding and his sharing of Indigenous Knowledges, as well as engagement with community.

Informing this approach was the postcolonial Indigenous theory perspective that seeks to build relationships between researchers and focus communities—while incorporating their historical and cultural relationship with the environment [33]. This integrates principles including relationality (between the environment, people, culture, and history), responsibility (of the researchers to be transparent and trustworthy in their engagement, portrayal, and data management), reverence (of ‘other ways of knowing’, and of respect and valuing of Indigenous Knowledges), and reflexivity (of understanding the researcher’s standpoint, influences, and previous paradigms for data comprehension) [33]. Aligned with this theory are the ‘CONSIDER reporting criteria’ for the reporting of ‘equitable research practices’, that ensure First Nations leadership of the research design and participant recruitment. The criteria also require authentic partnerships between First Nations and non-Indigenous allies [23]. These perspectives inform the paper structure to deliver policy-relevant findings that privilege First Nations Peoples’ input and relevance; this ideally moves responses from ‘culturally responsive policy’ to ‘decolonised and sovereign policy’ [34].

A narrative review was selected as the method. This approach seeks to access and consider a wide range of publication sources and content, including and in addition to academic publications, in order to describe this interdisciplinary topic [35]. Publications were reviewed from academic peer-reviewed literature as well as from government and non-government ‘grey literature’ in recognition of the limited range of remote community-specific climate change academic publications [17]. References were selected following the narrative review approach of Dhimal et al. [36], who detailed the value of PubMed for public health and biomedical publications, and the wide-ranging search capacity for open access publications of Google Scholar [36]. Academic publications were sourced using a keyword search in these databases (“climate change”, “Indigenous”, “Aboriginal”, “remote”, “Australia”, “Torres Strait”, “health”). Manual searching with these keywords was also conducted as expert guidance sought for recommended publications to ensure all recent and relevant publications were identified and included [37].

The focus on the Torres Strait Islands was selected to enable a closer investigation of the on-ground impact of the changes and its impacts on the population’s health and to gain from the cultural knowledge and leadership of co-author, Francis Nona. This focus enabled a holistic review of a case in a ‘natural’ setting—where the issues of concern are considered in their ‘reality’ [38]. Focusing on the health of Torres Strait Islander Peoples under climate change provides a perspective on the ‘lived experience’, where the ‘big picture’ of climate change is grounded in a ‘stark’ reality [1]. The further value of this focused approach is that the findings can often be explored and tested in other contexts beyond the Torres Strait region [39].

## 3. Results and Discussion

### 3.1. Community Focus: The Torres Strait Islands in a Changing Climate


*The Committee … recalls that the obligation of States parties [Australia] to respect and ensure the right to life extends to reasonably foreseeable threats and life-threatening situations that can result in loss of life.*



*The Committee considers that such threats [to the Torres Strait Islands and Islanders] may include adverse climate change impacts, and recalls that environmental degradation, climate change and unsustainable development constitute some of the most pressing and serious threats to the ability of present and future generations to enjoy the right to life.*
([40], s8.3)

The above statement was made by the UN Human Rights Committee in their September 2022 response to a claim by 14 Torres Strait Islanders (eight adults and six of their children) that Australia had ‘failed to implement an adaptation programme to ensure the long-term habitability of the islands’ ([40], s2.8). In terms of their health, the claimants listed concerns from increased disease and heat-related illness ([40], s2.6) and the ecological health of marine environments that support their nutrition ([40], s3.5). The claim was based on five articles under the UN International Covenant on Civil and Political Rights (2, 6, 17, 24 and 27) [40]. These articles cover respect and provision of rights by States to all individuals within its jurisdiction (article 2); protection of the inherent right to life (article 6); protection from interference with one’s privacy, family, or home (article 17); protection of minors by one’s family, society, and the State (article 24); and protection of minority groups’ culture and language (article 27) [41].

The overarching decision by the UN was that Australia had violated the claimants’ rights under the Covenant by ‘failing to implement adequate mitigation and adaptation measures to prevent negative climate change impacts on the authors and the islands where they live’ ([40], s8.2). Specifically regarding health, the UN Committee upheld that environmental harm can constitute a violation of human rights given the holistic independent nature of environment and culture, stating ‘the strong cultural and spiritual link between indigenous peoples and their traditional lands’ ([40], s5.7) and ‘the health of their land and the surrounding seas are closely linked to their cultural integrity’ ([40], s8.14). The Committee acknowledged that their decision establishes an international precedent for individuals impacted by climate change to lodge human rights claims against their national governments [42].

The claim to the UN by the 14 Torres Strait Islanders drew from the claimants’ personal experiences and from data that have documented unseasonal changes and extreme weather events—with associated impacts on human health. This includes direct health impacts from extreme weather events and heat, and indirect health impacts through changes to the physical environment, such as drinking water contamination, and to social structures, such as food supply chains [17,43]. The exposure to climate-related sea level rise and associated damage to Country, changing conditions for infectious disease, increased frequency and severity of cyclones, and disruptions to food, energy, and water security have been documented by organisations with local representation [44,45] and described in academic publications [2,3,46]. For example, some health impacts associated with climate change for this region were described in recent medically focused research:


*Climate-sensitive infections pose a disproportionate burden and ongoing risk to Torres Strait Islander peoples’ …. [for example,] the region constitutes 0.52% of Queensland’s population but has a disproportionately high proportion of the state’s cases: 20.4% of melioidosis.*
([2], p.1)

Beyond documented illnesses associated with climate change, there can be wellbeing impacts from damage to First Nations’ cultural heritage. This has been documented in a range of countries including Australia and New Zealand [47,48]. It was locally contextualised by Francis Nona when reflecting on his Torres Straits homeland:


*[Cultural damage] can already be seen in a number of inundations … and the [associated] distress caused by the damage to the cemetery. Areas containing rain stones or cultural significant trees, for example, are locations likely to remain unrecognised as important by Western scientists, but very significant to the people of the Torres Strait … [There will be a] loss of culture based on the climate changing.*
[1]

These impacts are documented in this article within a climate justice framework that recognises the causes of climate pollution as being unequally contributed by wealthier nations and peoples, while the impacts are experienced unequally by populations who did not necessarily contribute to the scale of the problem [49]. Otherwise known as ‘justice in the era of climate change’, this framework considers the ethics of actions into the future, appropriate emission reduction contributions and targets for nations, and ‘just transitions’ of communities, sectors, and peoples in adaptation responses [50]. Climate justice in the Torres Strait was described by Francis Nona in the context of impacts on culture:


*First Nations People, we oversee so much land, but we have impacted so little on the actual [causes] of climate change. We are not impacting, but yet we are experiencing—so what is that doing to culture?*
[1]

### 3.2. Core Values to Guide Culturally Appropriate Climate Change Mitigations and Adaptation

Responding to climate change involves both mitigation (to reduce emissions of greenhouse gases that create the conditions of change) and adaptation (to create resilience in populations and structures to cope with a changing climate) [51]. The Intergovernmental Panel on Climate Change (IPCC) proposed in 2022 that both these responses need to be undertaken in combination, given the interdependence of both approaches and the urgency and scale of climate action required. This approach has been coined ‘Climate Resilient Development’ (CRD) [22].

The process of the narrative review and the focus on climate change and health in the Torres Strait resulted in the identification of four core values that are critical to developing long-term and sustainable initiatives that constitute mitigation, adaptation, and/or in combination as CRD. These are support for self-determination [4], co-design of climate mitigation and adaptation initiatives to protect health [52,53], appropriate governance of initiatives [9,24,25], and respectful incorporation of Indigenous Knowledges [54,55]. Each is described below in the context of First Nations-led climate action responses and then specifically in the context of the Torres Strait Islands.

#### 3.2.1. #1: Self-Determination

Self-determination is defined by Australia’s National Health and Medical Research Council (NHMRC) as the ‘freedom to live well and live according to their values and beliefs’ ([25], p. 9). Self-determination is relevant to climate change response priorities where choice and control over responses are essential [25]. This is in contrast to some past research where First Nations Peoples have been ‘over-researched’ and data gathering has not always occurred in a transparent, respectful, or collaborative manner [5,24]. In seeking to value First Nations Peoples’ ways of knowing and cultural knowledges, self-determination is a core value. It supports empowerment of First Nations communities and agencies to plan and direct their own initiatives to adapt to the impacts of climate change, drawing from local leadership, local priorities, and the value of lived experience [9,27].

In the context of the Torres Strait, the *Meriba buay* community of practice for Torres Strait Islander health and wellbeing described the importance of self-determination in responding to environmental and social challenges, stating that:


*Research and experiential knowledge (i.e., personal knowledge, traditional knowledge, cultural knowledge) need to be mobilised and made more accessible to support the empowerment of Torres Strait Islanders to develop solutions to complex environmental and social problems’.*
[56]

This practice was communicated to a majority non-Indigenous audience of researchers by Francis Nona in a presentation where he explained:


*I just want to say as an Indigenous man please ensure that when you do work, you include Indigenous voices when working on solutions for complex health problems that affect us. Please ensure that this is just not a tokenistic measure but an authentic and significant contribution alongside many fields of work and groups of people involved in finding answers and options.*



*‘Nothing about us without us’ was a phrase coined in the context of patient centred care. It is just as valid when exploring issues that affect vulnerable groups being partners—not just subjects.*
[1]

#### 3.2.2. #2: Co-Design

The most recent IPCC Assessment Report explicitly described the importance of centring and valuing First Nations Peoples’ involvement in climate change responses in Australasia, stating:


*There is a central role for Indigenous Peoples in climate change decision making that helps address the enduring legacy of colonisation through building opportunities based on Indigenous governance regimes, cultural practices to care for land and water, and intergenerational perspectives.*
([29], p. 69)

This approach reflects the Australian 2022 Close the Gap in Indigenous Equity report that emphasises how First Nations-led initiatives and solutions are crucial to health responses [57]. Such involvement can be achieved through co-design, defined in the context of health within the Queensland Aboriginal and Torres Strait Islander Health Equity Framework as ‘the strongest type of partnership arrangement because it involves sharing decision -making authority’ ([53], p. 23) and, in doing so, seeks to hear and include First Nations’ ‘voices, lived experiences and cultural authority’ ([53], p. 23). Such partnerships can be developed through experience-based co-design, a methodology that seeks to reduce power imbalances between the investigators and participants [52].

Torres Strait Islander researchers have called for participatory approaches, including co-design and co-production of knowledge, for health-related climate responses to ‘assist Torres Strait Islanders to create their own meaning of health and find creative solutions to achieve positive health outcomes’ ([27], p. 178). Francis Nona has recommended co-design to effectively implement the Health Equity Framework in terms of climate change in the Torres Strait:


*In this initial engagement I think what is being seen is First Nations People having the understanding that it’s something for them, not about them … ultimately what I want to see moving forward is something for them as we work with them, not on them.*
[1]

#### 3.2.3. #3: Governance

The systems and processes within which power is shared, decisions are made, and responsibilities and roles of decision makers are defined and held can constitute Indigenous governance [58]. In responding to climate change impacts on the health of First Nations Peoples, Indigenous governance involves appropriate Traditional Owner permissions, community engagement, decision-making processes, cultural competence and safety, and following First Nations-developed protocols, including those regarding Indigenous Data Sovereignty [59] and those specifically focused on the Torres Strait [27,56,60]. The appropriate governance for responding to climate change and health risks in the Torres Strait was described by Francis Nona as:


*… led by the communities and people of the Torres Strait in respectful partnerships. These partnerships must include strong governance that addresses the power imbalances that are real between policy makers and community.*
[61]

#### 3.2.4. #4: Indigenous Knowledges

Indigenous Knowledges have been described by the UN as detailed understandings of local ecosystems and the changes within these—that is, being enriched, adapted, and passed down over generations [62]. First Nations’ Peoples bring a long history of lived experience and close environmental awareness about their local environment and how to adapt to climatic changes [15,22]. In Australia, oral traditions retain Indigenous Knowledges; these include stories of changing sea levels and associated responses from 7000 years ago and using indicator species to identify seasonal changes [54,55].

Previously, there has been stated resistance by non-Indigenous, Western science-trained researchers to Indigenous Knowledges being included in climate change research and information, such as the IPCC reports, until 2022 [15,22]. This has shifted recently; the latest IPCC Assessment Report 6 acknowledges that the Indigenous Knowledges of First Nations Australians can contribute to ecologically appropriate fire management, cultural water flows, and conservation plans, and thus ‘enhance effective adaptation through the passing down of knowledge about climate change planning that promotes collective action and mutual support’ ([29], p. 6). Future IPCC reports have been strongly encouraged to ensure First Nations author inclusion [21].

In the Torres Strait, a First Nations-led initiative recommends the value of integrating Indigenous Knowledges and those from other sources and disciplines to enable sufficient understanding and support local decision making—including about health [27]. Francis Nona’s work has also described the value of combining Western scientific and Indigenous Knowledges, stating:


*Scientific knowledge is important, but not more important than what our Elders and ancestors have taught us. We need the two ways of knowing to work together. To adapt to climate change, there needs to be both ways of knowing—using the Torres Strait knowledges in conjunction with western ways. There needs to be mutual respect on both sides.*
[61]

### 3.3. The Four Core Values in Action in the Torres Strait Climate Responses

Many of these four core values are inherently familiar to Torres Strait Islander Peoples. This was exemplified during a workshop convened in June 2022 to identify climate change adaptation and resilience priorities for the region. Over fifty Traditional Owners; government officers from local, state, and federal agencies; researchers; and civil society representatives participated, including the authors of this paper. Together, workshop participants documented statements that resonated with these four core values, notably:Climate change already affects daily lives in the region.Urgent and local climate action is required as soon as possible.Torres Strait Islander Peoples want to guide their region’s future.The Torres Strait region will become a leader in Indigenous climate adaptation and support a positive narrative and approach in responding to this challenge ([63], p. 3).

Potentially in response to this call and to the UN decision on the Torres Strait 8’s claim, the Australian Government’s latest budget committed $15.9 million for engaging First Nations Peoples on Climate Change [64]. This includes a proposed ‘Torres Strait Climate Centre of Excellence, enabling a coordinated regional response to better prepare for the impacts of climate change across the region’ ([65], p. 3).

## 4. Conclusions

This research identified then described four core values that could enhance the health and wellbeing of First Nations Australians in a changing climate. The Torres Strait region formed the focus of this paper, given both the Country connections of one of the co-authors, Francis Nona, and his professional work towards protecting the health and wellbeing of Torres Strait Islanders and their Country. He described his motivation and connection as:


*My ancestors have lived in this region for millennia and my wellbeing and that of my son is tied to an ongoing connection to this place … Ongoing climate change threatens to permanently damage parts of my homeland forever.*
[1]

This research was conducted with an intentional postcolonial Indigenous theory perspective to build relationships between First Nations and non-Indigenous researchers and to bring together different Knowledges. It considered aspects of climate justice in the human-induced causes of climate change and the political power dynamics that influence subsequent mitigation and adaptation responses.

The findings of this research were that health-protective climate responses appropriate to Torres Strait Islander Peoples and their Country need to embed the core values of self-determination, co-design of responses, culturally appropriate governance of initiatives, and incorporation of Indigenous Knowledges. These are likely to be relevant to other remote settings where First Nations Peoples live on their traditional Country. As the Australian Government moves forward with the funded climate initiatives to engage First Nations Peoples—including for the Torres Strait Climate Centre of Excellence—and its new Climate Change Bill, these core values can provide direction.

The words of Francis Nona in a presentation on his Country provide an apt summary to this research; he reminded fellow Torres Strait Islanders of the strengths that they bring to the climate change challenge:


*Strong, cohesive communities can work together to address issues that impact our community… Climate change is such a massive problem that it is hard to get motivated, but we need to acknowledge the strengths of our people in the Torres Strait and draw on this strength to develop local, Torres Strait Islander -led planning to address the impact climate change is, and will continue to have, on our homelands.*
[61]

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
