# Peer review of "Community-Led Health Initiatives for Torres Straits Island Communities in a Changing Climate: Implementing Core Values for Mitigation and Adaptation"

_ijerph, 2022, doi:10.3390/ijerph192416574_

Round 1
Reviewer 1 Report
This is a carefully considered, logically argued and very well presented paper. It specifically captures the values-based, power-sharing and local approach strongly preferred by Indigenous Peoples, including the Māori communities of Aotearoa New Zealand that I belong to and work with on these issues. In doing so the paper focusses on those directly impacted and the solutions they can offer, effectively transferring the focus away from the traditionally rules-based, controlling and power-driven approaches of English-derived scientific approaches. Those approaches have been unable to remedy the damage caused by the climate crisis, including the specific health issues set out in this paper, and need all the help they can get.
The paper encourages western scientists to recognise and understand that while their knowledge can be valued and should contribute to finding solutions to the climate crisis, it falls well short of being all-knowing. The paper provides a very diplomatic, yet clear and firm argument for the huge benefits and extensive data-informed understanding that Indigenous Knowledge systems can bring to western scientific data bases and methods. For Torres Strait Islanders, such knowledge systems have been honed and perfected over many millenia and are much older and more robust than the western scientific method which is still unable to address the climate crisis that is of its own making. Yet Indigenous Knowledge recognises that the knowledge that western science has, as this paper clearly argues, is essential to being able to fix the damage, provided it is applied using the four underlying values that are the focus of this paper.
This is an excellent and very important contribution to this field of research.
Author Response
Dear editors and reviewers,
We value your close reading and clear feedback on our submitted article.
In the attached, we have set out the reviewer’s reports and responded to each of the points requiring consideration. Where possible, we have revised our text to incorporate the reviewer’s comment.
Please let us know if further information is required to progress towards acceptance and publication.
Kind regards,
Nina Lansbury, on behalf of the author team

Reviewer 2 Report
This is a good paper with a worthy narrative. I have two major comments:
1 While the focus on Indigenous/First Nations community and the postcolonial Indigenous theory was strong, the positioning of the authors was not explicit. This reviewer has assumed that the first two authors do not identify as Indigenous/First Nations people, but the third author does. It is noted in the paper that one of the co-authors identifies/is from an Indigenous community, and has the links with the community that has initiated this work. Author order is significant, and indicates the power and structure of the project. If my assumptions are correct, and the Indigenous author is third in the list, and not the lead author of this paper, the colonial structure of the Academy is still in play. The paper led by Tania Huria (Huria, T., Palmer, S. C., Pitama, S., Beckert, L., Lacey, C., Ewen, S., & Smith, L. T. (2019). Consolidated criteria for strengthening reporting of health research involving indigenous peoples: the CONSIDER statement. BMC medical research methodology, 19(1), 173.) may be useful to make the positioning of the authors explicit, if my assumptions are incorrect and/or author reordering is not possible.
2 I got a little lost/found the middle paragraphs of the introduction section wordy. It may help if each or some of the sentences where 'documentation' or generic statements are referenced, that there is a little more specific comment from these papers included.
A minor editing comment is that there are (nearly) consistent missing spaces before hyphens when the hyphen is used as a pause for a list or example, such as line 11 (Peoples- especially, should be Peoples - especially), line 13 (Country- unlike, should be Country - unlike) and so on.
Author Response

(The authors gave the same response as above.)

Round 2
Reviewer 2 Report
The authors are to be congratualted for their responses and this paper.